# Variable Admittance Control Based on Human–Robot Collaboration Observer Using Frequency Analysis for Sensitive and Safe Interaction

**DOI:** 10.3390/s21051899

**Published:** 2021-03-08

**Authors:** Hyomin Kim, Woosung Yang

**Affiliations:** School of Robotics, Kwangwoon University, 20 Kwangwoon-ro, Nowon-gu, Seoul 01897, Korea; nankimhm@kw.ac.kr

**Keywords:** human–robot collaboration, physical human–robot interaction, admittance control

## Abstract

A collaborative robot should be sensitive to the user intention while maintaining safe interaction during tasks such as hand guiding. Observers based on the discrete Fourier transform have been studied to distinguish between the low-frequency motion elicited by the operator and high-frequency behavior resulting from system instability and disturbances. However, the discrete Fourier transform requires an excessively long sampling time. We propose a human–robot collaboration observer based on an infinite impulse response filter to increase the intention recognition speed. By using this observer, we also propose a variable admittance controller to ensure safe collaboration. The recognition speed of the human–robot collaboration observer is 0.29 s, being 3.5 times faster than frequency analysis based on the discrete Fourier transform. The performance of the variable admittance controller and its improved recognition speed are experimentally verified on a two-degrees-of-freedom manipulator. We confirm that the improved recognition speed of the proposed human–robot collaboration observer allows us to timely recover from unsafe to safe collaboration.

## 1. Introduction

Collaborative robotics has become the new frontier for industrial robots by combining high-level motion accuracy and the repeatability of robots with the flexible cognitive judgment of humans [1,2]. Effective human–robot collaboration requires an intuitive user interface to maximize operation flexibility [3]. The hand guiding collaboration mode, also known as direct teaching, is a representative collaboration mode defined in standards ISO 10218-1/2 [4,5]. In this mode, the operator directly sets the sequence of desired robot positions by moving the robot end effector without an intermediate interface. In addition to the intuitive interaction, the operator can manipulate the robot while receiving haptic feedback that guides or limits the trajectory. Thus, this mode evolves the interface bottleneck of traditional input devices such as mouse, keyboard, and joystick [6]. Nevertheless, when the operator and the robot are in continuous contact, safety during the physical human–robot interaction is the most important consideration [7]. Therefore, the robot must operate according to the operator intention while ensuring safety.

To ensure the operator’s safety while collaborating with a robot, standards ISO 10218-1/2 define a power and force limiting mode. They also prescribe managing the system’s output force within 0.5 s to prevent the plastic deformation of human skin during collisions [8]. In this mode, an energy tank-based observer is commonly used to limit the motor power and force of the industrial robot [9,10]. The energy tank-base provides an easy solution to the problem of passivity but requires an analysis of the total energy. This control prevents unintentional robot movement due to unexpected contact with the external environment. However, in the hand guiding mode, the external environment means that the operator, and it is difficult to generalize operator’s energy. As the physical properties of humans vary between individuals and human intention changes in real time. Moreover, energy tank-based control only considers the total energy. This can desensitize the system’s response by limiting the energy that increases with a sudden change in the operator’s intention, even if the system is stable [11]. Hence, in the hand guiding mode, for the safety and sensitivity control, the robot system should distinguish between the operator’s intention and the unstable system behavior.

To identify the operator’s intended behavior, we propose an observer based on frequency analysis. The upper limb motion of a human can present frequency components within 5 Hz [12], and voluntary motion has components below 2 Hz [13,14]. Therefore, a safe human–robot collaboration state should have a frequency below 2 Hz, whereas an unsafe human–robot collaboration state can be identified by frequency components between 2 and 5 Hz. Stability analysis methods based on the frequency analysis of systems such as this paper with various types of input signals and applications have been studied. Ryu et al. proposed a haptic stability observer (HSO) that is based on the position-frequency analysis of the robot end effector and quantifies the degree of system instability [15]. Dimeas and Aspragathos proposed an instability observer based on the frequency analysis of force signals generated by the human–robot interaction [16]. Okunev et al. used a trained frequency domain classifier based on AdaBoost and developed a control strategy to prevent the oscillation of the robot manipulator [17]. These studies used discrete Fourier transform (DFT) for frequency analysis. The DFT is commonly used for frequency analysis but has a tradeoff between the sampling time and frequency resolution. For instance, a DFT with 3.91 Hz resolution is obtained from 256 samples at a 1 ms period. In this case, DFT can analyze the signal within 0.5 s, but it is difficult to clearly recognize the operating frequency within 5 Hz. Even if the DFT has a 1 Hz resolution, it would require a long window of 1024 samples. However, this window size makes the observer to hindering a quick response when unsafe interaction occurs.

The main contribution of this study is the development of an alternative algorithm that overcomes the limitations in computational speed and resolution of a DFT-based observer. We propose a human–robot collaboration observer (HRCO) based on an infinite impulse response (IIR) Butterworth filter. In addition, this observer applies the variable admittance controller. The variable controller makes it possible to configure a sensitive and safe system by setting low admittance parameters in a safe collaboration state and high admittance parameters in an unsafe collaboration state [18]. We verify that the proposed observer can recognize the unsafe collaboration state within 0.5 s and can allow a full recovery to the safe collaboration state.

The rest of the paper is organized as follows. Section 2 introduces the characteristics and limitations of the admittance control. In Section 3 we introduce the HRCO based on an IIR filter. Then, we propose the variable admittance control based on the HRCO for sensitive and safe human–robot collaboration. In Section 4, we compare the performance of HRCO and HSO through the simulation verification. In Section 5, the proposed HRCO and controller are applied to a two-degree-of-freedom (DOF) manipulator to verify its performance. Finally, we discuss the results and provide directions of future work in Section 6.

## 2. The Admittance Control Scheme

Admittance control and impedance control are generally used for force control according to the operator intention [19,20]. They adjust the end effector based on a virtual model with a desired inertia and damper. Admittance control can easily become unstable under sudden changes in external impedance [21]. On the other hand, impedance control can easily become unstable under low impedance. Impedance control is robust to sudden changes in environmental stiffness [22,23]. Therefore, in this study, admittance control is used to easily cause an unsafe collaboration state.

The admittance model defines the robot behavior in free space and contact space by dividing the robot state into before and after contact with the external environment [24]. The admittance equations in free space are given by
(1)$$f_{\mathrm{ext}}=m_{d}\left({\ddot{x}}_{r}-{\ddot{x}}_{0}\right)+d_{d}\left({\dot{x}}_{r}-{\dot{x}}_{0}\right)+k_{d}\left(x_{r}-x_{0}\right),$$
(2)$$f_{\mathrm{ext}}=f_{h}-f_{\mathrm{vir}},$$
where $x_{r}$ is the position of the one-DOF robot model, $x_{0}$ is the target position, $m_{d}$, $d_{d}$, and $k_{d}$ are admittance parameters, namely, desired inertia, damper, and stiffness, respectively, $f_{\mathrm{ext}}$ is the external force applied to the robot given in Equation (2), $f_{h}$ is the interaction force between the operator and robot, and the virtual force $f_{\mathrm{vir}}$ is a feedback force which can be a guiding force during collaboration or a virtual force in a haptic system.

During human–robot interaction, the operator’s hand is always in contact with the end effector of the manipulator. Therefore, the admittance model in the contact space should be applied as follows for $x_{0}$, ${\dot{x}}_{0}$, ${\ddot{x}}_{0}$, and $k_{d}$ being set to zero [25]:(3)$$f_{\mathrm{ext}}=m_{d}{\ddot{x}}_{r}+d_{d}{\dot{x}}_{r},$$
(4)$${\ddot{x}}_{d}=\left(f_{\mathrm{ext}}-d_{d}{\dot{x}}_{r}\right)/m_{d}.$$

Equation (4) provides the desired acceleration according to the applied external force in Equation (3). In general, admittance control includes an inner closed-loop position controller to follow the desired position $x_{d}$ integrated by ${\ddot{x}}_{d}$.

Figure 1 shows the block diagram of the admittance control used for stability analysis. Control includes the admittance model $C_{1}$, inner position controller $C_{2}$, one-DOF model $G_{p}$, human impedance $Z_{h}$, and linear first-order time delay system $H_{d}$. In the figure, $m_{r}$ is the mass of the one-DOF robot, $k_{h}$ is the human stiffness, $t_{d}$ is the delay time, and $k_{p}$ and $k_{v}$ are the proportional and derivative control gains of the inner position controller, respectively.

The transfer function for admittance control obtained from Figure 1 is given by
(5)$$C_{\mathrm{adm}}\left(s\right)=\frac{x_{r}\left(s\right)}{{\hat{f}}_{\mathrm{ext}}\left(s\right)}=\frac{C_{1}C_{2}G_{p}+G_{p}}{1+C_{2}G_{p}},$$
where $m_{r}$ and $t_{d}$ were set to 3 kg and 1 ms in this study. In addition, gains $k_{p}$ and $k_{v}$ were set to 100 N/m and 20 Ns/m, respectively, to obtain a critically damped system, and $k_{h}$ was set to 176.39 N/m based on experimental measurements of human stiffness in [26]. In addition, $f_{\mathrm{vir}}$ was set to 0 N for the stability analysis. Finally, ${\hat{f}}_{\mathrm{ext}}$ was the external force through the time delay function $H_{d}$.

Admittance controller configured as shown in Equation (5) has different control characteristics according to the setting of admittance parameters such as desired inertia and damper. In human–robot interaction, the robot is set sensitively to reduce the operator’s effort [15,16]. If admittance parameters such are set with low gains at a specific ratio [16], the robots are sensitive to the operator intention. However, such low-value parameters may render the admittance controller unstable when the operator intention changes rapidly or a collision with a rigid object occurs [16,23]. In contrast, high values for the admittance parameters can maintain stable control even under environmental disturbances. However, high values demand high interaction forces to move the robot. Therefore, the optimal admittance parameters should be dynamically determined to achieve both sensitivity and safety.

## 3. Variable Admittance Control Based on HRCO

This section proposes a control strategy for recovering from an unsafe collaboration state to a safe collaboration state using variable admittance control based on HRCO. To overcome the limitations of fixed admittance parameters and enable a suitable response to unknown environmental disturbances, variable admittance control has been devised [27,28]. This strategy establishes model-free control with online adjustment of the admittance parameters. Variable admittance control allows the configuration of a sensitive and safe system by setting low admittance parameters in a safe collaboration state and high admittance parameters when safety is compromised. To leverage these variable characteristics, we need an observer that accurately identifies the collaboration condition to adjust the admittance parameters.

### 3.1. The Human–Robot Collaboration Observer

The proposed observer uses an IIR Butterworth filter to overcome the limitations of the DFT-based stability observer. The input signal uses the 2 × 1 external force vector $F_{\mathrm{ext}}$ instead of the end-effector position because the output position is less sensitive than the input force signal due to the low-pass filter characteristics of the admittance model [16].

The IIR Butterworth digital filter performs recursive computations using previous input and output signals. Additionally, this filter provides an optimal Taylor series approximation of the ideal filter response at analog frequencies [29]. The block diagram of the IIR Butterworth filter is shown in Figure 2a, where u is the input signal, y is the output signal, P and Q are the filter order of feedforward and feedback filters, respectively, a and b are the coefficients of the feedforward and feedback filters, respectively, and $Z^{-1}$ is the unit delay. The frequency resolution is determined by an analog-to-digital converter. The difference equation of the IIR filter is given by
(6)$$y\left(n\right)=\frac{1}{a_{0}}\left(\sum_{i=0}^{P}b_{i}u\left(n-i\right)-\sum_{j=1}^{Q}a_{j}y\left(n-j\right)\right).$$

From Equation (6), we construct a low-pass filter (LPF) and a high-pass filter (HPF) with P and Q being set to the second order. In addition, the cutoff frequency $\omega_{c}$ is 5 Hz considering the operating frequency of the human upper limb, and the sampling period is 1 ms. The detailed parameters configuring these filters are shown in Table 1.

Figure 2b shows the magnitude response of the filters according to Table 1. As the input frequency increases, the magnitude response of the LPF is decreased, and the HPF is increased. As the IIR Butterworth filter is used, the LPF and HPF respond monotonically [30]. Further, even if the input signal’s magnitude is changed, the ratio between LPF and HPF in each frequency component is constant, and the same magnitude is obtained at the cutoff frequency. Figure 2c is a pole-zero plot for evaluating the stability of a discrete system such as the IIR digital filter. The zeros of the LPF are located at −1 to attenuate the high-frequency signal, and the zeros of the HPF are located at 1 to attenuate the low-frequency signal. The poles of both filters have the same position and are in the unit circle, confirming that they are designed to be stable.

Considering the characteristics of the IIR Butterworth filters, we propose an observer to analyze the input frequency while disregarding the magnitude of the input signal:(7)$$I_{O}^{n}=\left\{\begin{array}{c} 0\\ \left\|F_{\mathrm{ext},H}^{n}\|/\|F_{\mathrm{ext},L}^{n}\right\| \end{array}\;\begin{array}{c} \|F_{\mathrm{ext},L}^{n}\|<0.01,\\ \mathrm{Otherwise}, \end{array}\right.$$
where $I_{O}$ is a dimensionless value between 0 and 1. The $\left\|F_{\mathrm{ext},L}^{n}\right\|$ and $\left\|F_{\mathrm{ext},H}^{n}\right\|$ represent the Euclidean norm of the *n*-DOF external force passing through the LPF and HPF, respectively. When $\left\|F_{\mathrm{ext},L}^{n}\right\|$ is below 0.01 N, the output of $I_{O}$ is set to zero to prevent an abrupt output variation due to division by a value very close to zero. Additionally, to be robust with the noise, we use a derivative filter to smooth $I_{O}$ by applying Equation (8), thus establishing the proposed HRCO with $I_{\mathrm{HRCO}}$ output as shown in Figure 3, which is a dimensionless value between 0 and 1. For the derivative filter in the proposed HRCO, η is set to 0.02.
(8)$$I_{\mathrm{HRCO}}^{n}=\left(1-\eta \right)I_{\mathrm{HRCO}}^{n-1}+\eta \left(I_{O}^{n}\right).$$

### 3.2. Admittance Adjustment Method Based on HRCO

Various studies have focused on maintaining system stability by adjusting the admittance parameters on-line. In [19,31], variable admittance control was proposed to obtain an overdamped system by increasing desired damper $d_{d}$ while maintaining desired inertia $m_{d}$. Such damping controllers are commonly used to stabilize the system. By only adjusting the damper, although the magnitude decreases, the vibration frequency is maintained. Thus, the resulting vibration can cause discomfort to the operator while moving the robot [17]. To prevent discomfort, this study uses variable admittance control with increasing $m_{d}$ and $d_{d}$ at a fixed ratio [16]. In addition, we analyze the control stability for a one-DOF linear dynamics model [23]. Although this type of stability analysis is common, it disregards model uncertainty and nonlinearity. Therefore, it can only provide a qualitative evaluation of the frequency response during human–robot collaboration [16].

Figure 4a shows the frequency response of the admittance control for human stiffness of 176.39 N/m, and Figure 4b shows the root locus plot of admittance control for increasing external stiffness. The red curve shows the results of admittance control that obtains a sensitive response for $m_{d}$ and $d_{d}$ set to 1 kg and 10 Ns/m, respectively. The green curve indicates the control result in which admittance parameters are increased five times at a fixed rate. In the root locus plot, the sensitively set admittance control is in the right plane when the external stiffness increases as much as the human stiffness. This means that when a person holds the robot handle, the controller becomes unstable. For this unsafe collaboration state, increasing $m_{d}$ and $d_{d}$ at a fixed ratio makes the controller asymptotically stable even for human stiffness, as seen Figure 4b. As admittance parameters increase, the magnitude decreases, and the phase shifts to the left, as shown in Figure 4a. This means that the natural magnitude and frequency decrease. Therefore, this control strategy is expected to effectively reduce both the vibration magnitude and frequency.

Using the characteristics of admittance control, we propose a method for adjusting the admittance parameters according to the human–robot collaboration state as follows:(9)$$m_{d}=\left\{\begin{array}{c} m_{d,0}\\ m_{d,0}+\alpha \left(I_{\mathrm{HRCO}}-I_{\mathrm{HRCO},0}\right) \end{array}\begin{array}{c} I_{\mathrm{HRCO}}<I_{\mathrm{HRCO},0},\\ I_{\mathrm{HRCO}}\geq I_{\mathrm{HRCO},0}, \end{array}\right.$$
(10)$$d_{d}=\frac{d_{d,0}}{m_{d,0}}m_{d}.$$
where $m_{d,0}$ and $d_{d,0}$ are initial values of $m_{d}$ and $d_{d}$. They are set sensitively to manipulate the robot with low interaction force. $I_{\mathrm{HRCO},0}$ is the value of $I_{\mathrm{HRCO}}$ at 2 Hz that distinguishes between safe and unsafe collaboration state. The α is the sensitivity weight for admittance adjustment. It makes the admittance parameters become insensitive in proportion to the operating frequency. Through this adjustment strategy, allows the robot to recover from unsafe to safe interaction.

## 4. Simulation Verification of HRCO

In this section, we compare the HRCO with HSO to evaluate the performance of the proposed observer. HSO is an online stability index that is based on frequency analysis using DFT [15]. It is the ratio of the sum of magnitudes at the unstable frequency range over the sum of magnitudes at the total frequency range, as in Equation (11).
(11)$$I_{\mathrm{HSO}}=\frac{\sum_{f_{\mathrm{unstable}}}H}{\sum_{f_{\mathrm{all}}}H}$$
where H is the magnitude of the signal for each frequency. In this study, to match the frequency region of interest, the HSO was set to analyze frequency signals in units of 1 Hz, as shown in Table 2. The unstable frequency range of HSO was set above 2 Hz and the input signal was also set as a force signal as in HRCO.

The first simulation evaluates the observer’s performance for an input signal that has various frequencies and sizes. An input signal is applied along the *x* axis for a frequency between 0 and 10 Hz in Figure 5a. The magnitude of the external force is set to 0.5 and 1 N, as shown in Figure 5b. A force signal is set to 16-bit resolution within ±40 N considering the performance of the analog-to-digital converter.

Figure 5c presents the result of HSO. The HSO has a value proportional to the input frequency, but does not completely go to zero at 20 s when the input signal is 0 Hz because it is sampled in a moving window involving 1024 data, which causes a time delay.

The gray curve in Figure 5d represents the $I_{O}$, which responds to the input frequency without being affected by the magnitude of the input signal. However, the $I_{O}$ has severe jitter that hinders its use as a control input. A derivative filter was used to smoothen $I_{O}$, resulting in the blue curve $I_{\mathrm{HRCO}}$. From the results in Figure 5d, it was confirmed that HRCO can analyze the frequency without using DFT.

In the second simulation, to guarantee fast recognition using the proposed HRCO, we analyzed the step input response to input signals of 1 N at frequencies of 1–5 Hz. The response of HSO to the step input signal occurs after about 1 s because it is sampled in the moving window for 1024 data, as shown in Figure 6a. This recognition speed is too slow to minimize human injuries in the event of an accident. Additionally, due to the low resolution, it is impossible to distinguish the signals over 2 Hz. Therefore, it is possible to distinguish the safe or unsafe collaborative state, but it is not suitable to adjust the admittance parameter in the unsafe collaborative state.

The HRCO can distinguish each frequency input from 1–5 Hz. More detailed, the frequency analysis resolution of HRCO depends on the performance of the IIR filter. It also depends on the analog-digital converter. However, it only affects the setting of the cutoff frequency resolution. IIR Butterworth filters have a monotonically and linearly changing magnitude function with the input frequency [30]. Hence, the proposed HRCO is linear with the input frequency, and its frequency resolution is almost infinite. Additionally, HRCO has a steady value of 0.16 for 2 Hz input signals, as shown in Figure 6b. This value is denoted as $I_{\mathrm{HRCO},0}$ and used to distinguish between safe and unsafe collaboration state. In addition, HRCO has a settling time of 0.29 s regardless of the input frequency, representing a recognition speed 3.5 times faster than that obtained using the DFT, whose delay is 1.024 s for a resolution of 0.98 Hz at a 1 ms sampling time. Therefore, the proposed controller based on the HRCO will be expected to reduce the impact force on humans in less than 0.5 s after detecting unsafe interaction.

## 5. Experimental Evaluations of Variable Admittance Control Based on HRCO

### 5.1. Experimental Setup

Figure 7 shows the two-DOF manipulator constructed to evaluate the proposed variable admittance controller based on the HRCO. The link lengths are 0.4 and 0.33 m, and their respective weights are 2.4 and 1.7 kg. The maximum torque generated by each joint is 32.5 Nm. The robot system was configured using TwinCAT 3.1 software from the Beckhoff Company (Verl, Germany) to guarantee a 1 ms control period.

The proposed variable admittance control based on HRCO includes the admittance model, HRCO, parameter adjustment, and inner position controller. The admittance model of the robot includes Equations (13) and (14) based on Equations (3) and (4). Vectors $X_{r}$ and $X_{d}$ of dimension 2 × 1 are the end-effector position and desired position, respectively. External force $F_{\mathrm{ext}}$, virtual force $F_{\mathrm{vir}}$, and human force $F_{h}$ are also represented by 2 × 1 vectors. The desired inertia $M_{d}$ and desired damper $D_{d}$ are 2 × 2 diagonal matrices. They are adjusted by the parameter adjustment block from Equations (9) and (10).
(12)$$F_{\mathrm{ext}}=F_{h}-F_{\mathrm{vir}}$$
(13)$$F_{\mathrm{ext}}=M_{d}{\ddot{X}}_{r}+D_{d}{\dot{X}}_{r}$$
(14)$${\ddot{X}}_{d}=M_{d}^{-1}\left(F_{\mathrm{ext}}-D_{d}{\dot{X}}_{r}\right)$$

The inner position controller for following the desired position $X_{d}$ is based on the computed torque method as follows:(15)$${\ddot{X}}_{\mathrm{ref}}={\ddot{X}}_{d}+K_{v}\left({\dot{X}}_{d}-{\dot{X}}_{r}\right)+K_{p}\left(X_{d}-X_{r}\right),$$
(16)$${\ddot{q}}_{\mathrm{ref}}=J^{-1}\left({\ddot{X}}_{\mathrm{ref}}-\dot{J}\dot{q}\right),$$
(17)$$\tau =M\left(q\right){\ddot{q}}_{\mathrm{ref}}+C\left(q,\dot{q}\right)+G\left(q\right)+J^{T}F_{h}.$$

Equation (15) corresponds to the error dynamics of the computed torque method, and reference acceleration ${\ddot{X}}_{\mathrm{ref}}$ is a 2 × 1 vector. Proportional gain $K_{p}$ and derivative gain $K_{v}$ of the inner position controller are, respectively, set to 10,000 N/m and 200 Ns/m to achieve robustness and obtain a critically damped system. In addition, J is the 2 × 2 Jacobian matrix, q and τ are 2 × 1 vectors of joint angles and output torques, respectively, $M\left(q\right)$ is the 2 × 2 inertia matrix, $C\left(q,\dot{q}\right)$ is the 2 × 1 Coriolis and centrifugal force vector, and $G\left(q\right)$ is the 2 × 1 gravitational torque vector.

To prevent the operator from being harmed during unsafe collaboration, human force $F_{h}$ is generated by the human impedance model:(18)$$F_{h}=M_{h}{\ddot{X}}_{h}+D_{h}{\dot{X}}_{h}+K_{h}\left(X_{h}-X_{r}\right),$$
where $X_{h}$ is a 2 × 1 vector that represents the virtual human position, $M_{h}$, $D_{h}$, and $K_{h}$ are 2 × 2 diagonal matrices of human inertia, damper, and stiffness set to 1.27 kg, 12.02 Ns/m and 176.39 N/m, respectively [26]. In addition, white noise with a standard deviation of 0.02 N is added considering the sensor noise of the force/torque sensor. The moving speed of the virtual human is set between 0.02 and 0.3 m/s, as established in [32] from experiments to measuring the speed of human movement in point-to-point operation.

A virtual surgical simulation environment was considered for the experiments. A virtual object has average stiffness and viscosity at the interface between a needle and a ligament. Thus, $K_{\mathrm{vir}}$ and $D_{\mathrm{vir}}$ were set to 300 N/m and 150 Ns/m, respectively [21]. For $X_{\mathrm{vir}}$, a 2 × 1 matrix that represents the virtual object position, the virtual force is generated as follows:(19)$$F_{\mathrm{vir}}=\left\{\begin{array}{cc} 0 & X_{\mathrm{vir}}>X_{r},\\ -D_{\mathrm{vir}}\dot{X}+K_{\mathrm{vir}}\left(X_{\mathrm{vir}}-X_{r}\right) & \mathrm{Otherwise}. \end{array}\right.$$

### 5.2. Experimental Results

Admittance control often becomes unstable when the impedance parameters change suddenly, such as when the operator moves at high speed or a collision with a rigid object occurs. Thus, to verify the effectiveness of the proposed variable admittance control to restore safe collaboration from an unsafe collaboration state, two experiments were conducted considering unstable operation. The environments for the two experiments are a sudden change in human impedance and a virtual rigid wall collision environment. The experimental parameters set for each experiment are shown in Table 3. In each experiment, we verified the performance of admittance control for low, high, and variable admittance parameters. The parameters of the controllers that are compared and evaluated in the experiment are shown in Table 4. All controllers $M_{d}$ and $D_{d}$ were set at a constant ratio of 10. The low and high admittance parameters were set to cause unsafe and safe collaboration, respectively. For variable admittance control, the initial values were set to the same as the low admittance parameters.

The first experiment considered a sudden change in human impedance. A virtual human started with an acceleration of 0.5 m/s^2^ at initial position $X_{\mathrm{init}}$ of −0.15 m, as shown in Figure 8a, and moved with a maximum speed of 0.3 m/s, as shown in Figure 8b. Then, the virtual human stopped at the final position $X_{\mathrm{fin}}$ of 0.15 m, as shown in Figure 8c. The deceleration of the virtual human was set to −10 m/s^2^ for generating a sudden change of human impedance at the stopping position. Figure 9 shows the experimental results over time, and Figure 10 shows the position–velocity graph, where the filled marker in each graph represents the final position of the corresponding experiment.

The red curve in Figure 9a indicates the end-effector position obtained from admittance control with low admittance parameters. As expected, this controller cannot handle a sudden change of the human impedance due to the high deceleration. Figure 10a shows the residual vibration. This vibration causes unsafe collaboration, as verified in Figure 9c.

Admittance control with high admittance parameters provides the lowest overshoot and no vibrations, indicated by the green curves in Figure 9a and Figure 10b. The HRCO under this controller is below $I_{\mathrm{HRCO},0}$, except for the moment when the virtual human suddenly stopped, as shown in Figure 9c. Thus, this controller maintains safe collaboration even after the sudden change in operator’s intention. However, this controller requires a continuous interaction force of approximately 5 N for hand guiding at 0.3 m/s, as shown in Figure 9b. Moreover, Figure 10b shows that this controller causes the largest position error in steady state among the compared controllers.

The blue curves in Figure 9 and Figure 10 illustrate the performance of the proposed variable admittance control based on the HRCO. As shown in Figure 9d, the admittance parameters are adjusted according to the HRCO output. As a result, a low interaction force of approximately 1 N is guaranteed during the motion, as shown in Figure 9b. This force is five times smaller than that required during control with the high admittance parameters. Figure 9a and Figure 10c show that when stopping with a large deceleration, the control limits residual vibrations to maintain the safe collaboration state despite the control overshoot.

The second experiment considered a sudden change in impedance due to the collision with a virtual rigid object. The virtual human moved 0.1 m along the *x* axis with a minimum speed of 0.02 m/s and collided with a virtual object located at 0.05 m (Figure 11a). The two-DOF manipulator was controlled according to the human force until the collision occurred, as shown in Figure 11b. After collision, the manipulator stopped at the equilibrium position of the human force and virtual force, as shown in Figure 11c. Figure 12 and Figure 13 show the experimental results obtained from the evaluated controllers for collision with a virtual object.

The results of admittance control with low admittance parameters are shown as the red curves in Figure 12 and Figure 13. This controller generates small vibrations due to joint friction during the quasi-static movement at 0.02 m/s, as shown in Figure 13a. This controller is not asymptotically stable after collision with the virtual object and presents a large oscillation. Consequently, the manipulator implementing this controller is likely to harm the operator. Moreover, the HRCO indicates unsafe collaboration, as shown in Figure 12c.

The controller with high admittance parameters maintains safe collaboration even after the collision, as shown in the green curves in Figure 12 and Figure 13. However, this controller is insensitive to the operator’s input, demanding a high control input for generating the quasi-static movement that overcomes the joint friction. As a result, the manipulator cannot be guided according to the target speed, as shown in Figure 13b. Furthermore, this controller presents a haptic feedback error of 3 N after the collision, as shown in the green curve in Figure 12b

The blue curves in Figure 12 and Figure 13 show the experimental results of the proposed variable admittance control. With sensitive initial admittance parameters, the proposed controller responds to small control inputs initially. Unlike the admittance controller with low admittance parameters that causes vibration during quasi-static movement, the proposed controller improves the operator low-speed motion intention by restraining the vibration, as shown in Figure 13c. After the collision, the admittance parameters are continuously adjusted to recover from an unsafe collaboration state to safe collaboration state. After collision, the proposed controller does not become unstable and shows a haptic feedback error close to 0 N, as shown in Figure 12b.

## 6. Conclusions

A robot interface that requires continuous contact between the operator and robot should distinguish between intended and unintended operator motions and respond sensitively while maintaining safety and stability. To this end, we propose the HRCO, an operation frequency analysis method using a second order IIR Butterworth filter instead of the conventional DFT. The HRCO can recognize the operator intention within 0.29 s, which is approximately 3.5-times faster than the DFT-based frequency analysis at a resolution of 1 Hz. In addition, the HRCO has a low computational cost compared with the DFT, which requires 1024 recursive operations. As verified experimentally, the HRCO is able to determine operation safety in real time.

We also propose a variable admittance controller based on the HRCO for stably controlling a robot interface for hand guiding according to the operator intention. The proposed controller restrained the response force after achieving the maximum interaction force, as shown in Figure 9b. Additionally, the admittance parameters were rapidly adjusted within 0.5 s after collision with a virtual object to prevent unsafe collaboration, as shown in Figure 12d.

The controller, control gain, and experimental environment evaluated in this study were set to trigger unsafe collaboration to verify the performance of the proposed HRCO. In general, admittance control and impedance control are used as force controllers based on the operator intention. Admittance control can easily become unstable under sudden changes in external impedance. On the other hand, impedance control can easily become unstable under low impedance. Thus, applying our weighted hybrid admittance–impedance controller proposed in [23] may allow us to maintain and recover safe collaboration. In addition, the proposed HRCO can be applied not only to a collaborative robot but to various robot systems that physically interact with operators, such as wearable robots and haptic systems. In future studies, HRCO will be applied as a cost function to multi-objective optimization, such as minimizing interactions and securing robustness against various external disturbances [33].

## Figures and Tables

**Figure 1 sensors-21-01899-f001:**
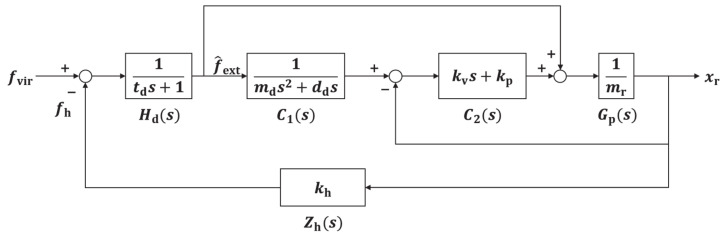
Block diagram of admittance control for stability analysis.

**Figure 2 sensors-21-01899-f002:**
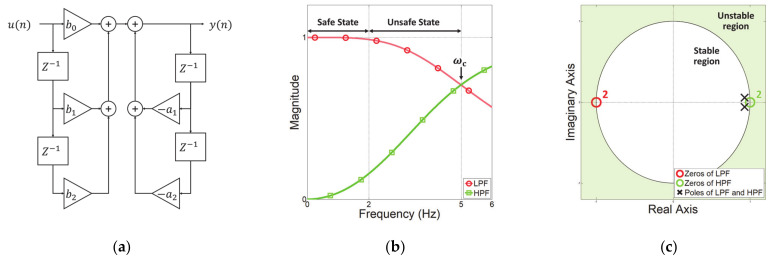
Design of the IIR Butterworth filter (**a**) Structure of 2nd IIR filter. (**b**) Magnitude response of the LPF and HPF. (**c**) Pole-zero map of LPF and HPF.

**Figure 3 sensors-21-01899-f003:**
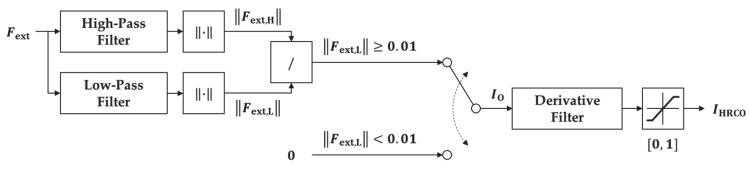
Block diagram of human–robot collaboration observer.

**Figure 4 sensors-21-01899-f004:**
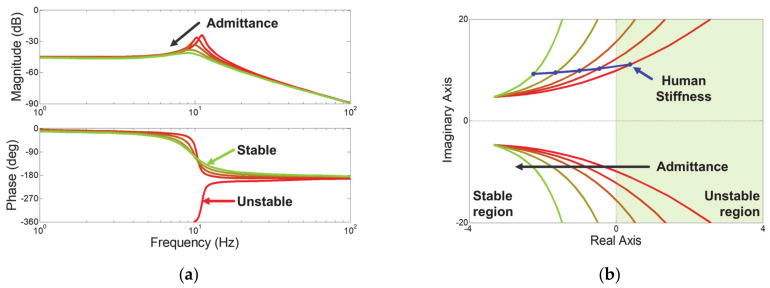
Stability analysis of admittance control for desired inertia $m_{d}$ and damper $d_{d}$ at a fixed ratio. (**a**) Frequency response for human stiffness of 176.39 N/m. (**b**) root locus plot for increasing external stiffness.

**Figure 5 sensors-21-01899-f005:**
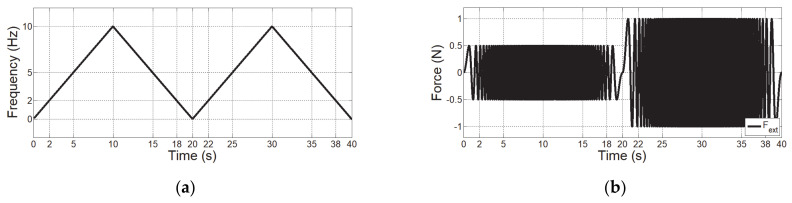
Simulation verification according to various magnitudes and frequencies. (**a**) Frequency of input force. (**b**) Magnitude of input force. (**c**) $I_{\mathrm{HSO}}$ output (red curve). (**d**) $I_{O}$ output (gray curve) and $I_{\mathrm{HRCO}}$ output (blue curve).

**Figure 6 sensors-21-01899-f006:**
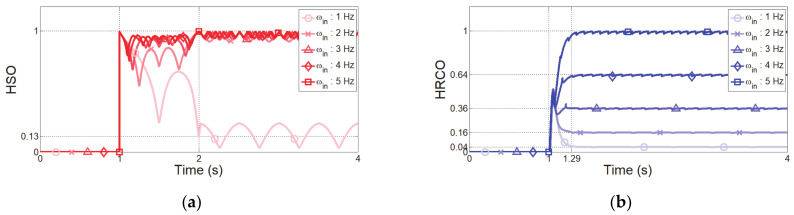
Step input response for input frequencies of 1–5 Hz. (**a**) $I_{\mathrm{HSO}}$
output (red curve). (**b**) $I_{\mathrm{HRCO}}$ output (blue curve).

**Figure 7 sensors-21-01899-f007:**
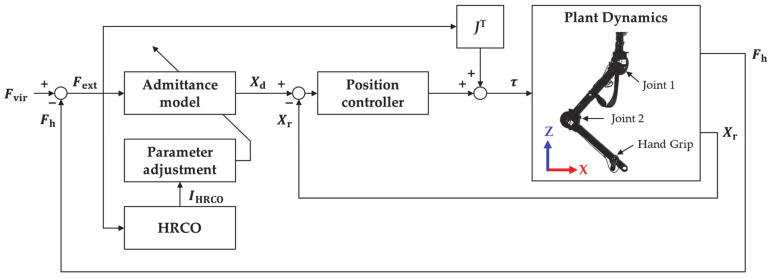
Block diagram of variable admittance control based on HRCO.

**Figure 8 sensors-21-01899-f008:**
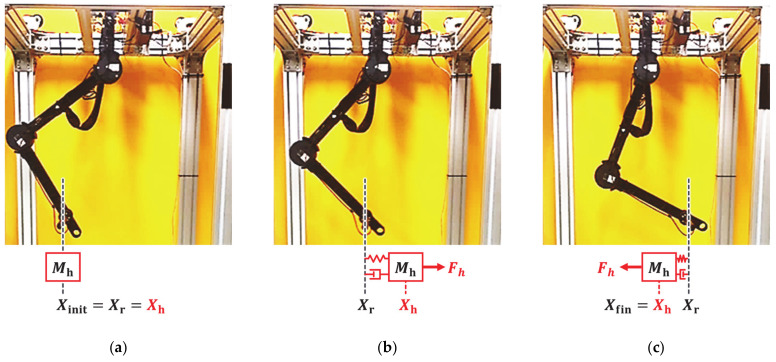
Experimental setup for sudden change of operator’s intention. (**a**) Starting position. (**b**) Motion with constant speed. (**c**) Stop with sudden deceleration.

**Figure 9 sensors-21-01899-f009:**
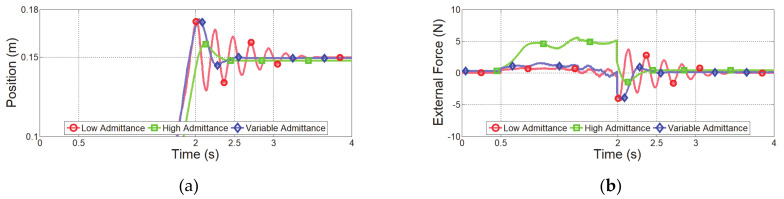
Experimental results of controllers under sudden change in operator’s intention. (**a**) End-effector position along *x* axis, (**b**) external force along *x* axis, (**c**) HRCO output, and (**d**) admittance parameters.

**Figure 10 sensors-21-01899-f010:**
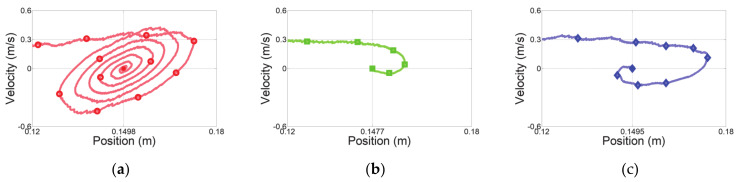
Position–velocity graphs along the *x* axis for experiment with sudden change in operator’s intention. Admittance control with (**a**) low and (**b**) high admittance parameters and (**c**) proposed variable admittance control based on HRCO.

**Figure 11 sensors-21-01899-f011:**
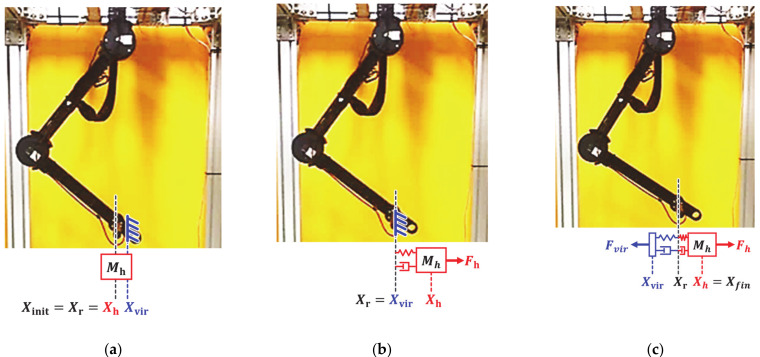
Experimental setup for virtual object collision. (**a**) Starting position. (**b**) Motion with constant speed. (**c**) Collision with virtual object.

**Figure 12 sensors-21-01899-f012:**
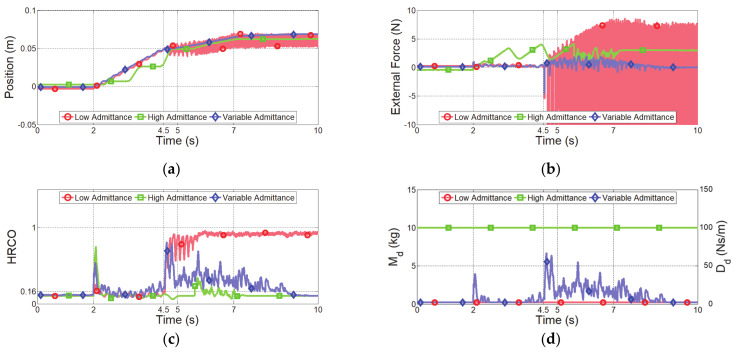
Experimental results of controllers for collision with virtual object. (**a**) End-effector position along *x* axis, (**b**) external force along *x* axis, (**c**) HRCO output, and (**d**) admittance parameters.

**Figure 13 sensors-21-01899-f013:**
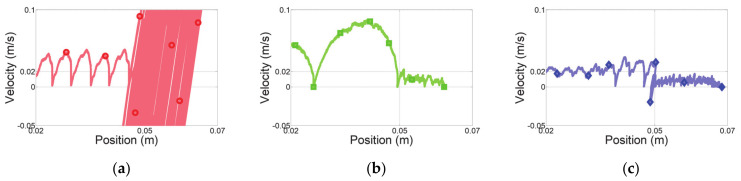
Position–velocity graphs along *x* axis for collision with virtual object. Admittance control with (**a**) low and (**b**) high admittance parameters and (**c**) proposed variable admittance control based on HRCO.

**Table 1 sensors-21-01899-t001:** Parameters of IIR low-pass and high-pass filters.

**Filter Type**	$a_{1}$	$a_{2}$	$b_{0}$	$b_{1}$	$b_{2}$
LPF	−1.9556	0.9565	0.000241	0.000483	0.000241
HPF	−1.9556	0.9565	0.978	−1.9561	0.978

**Table 2 sensors-21-01899-t002:** Parameters of HSO and HRCO.

	HSO	HRCO
Frequency Analysis Algorithm	DFT	IIR filter
Sampling frequency	1000 Hz	1000 Hz
Number of sampling data	1024 ea	2 ea
Frequency resolution	0.98 Hz	∞

**Table 3 sensors-21-01899-t003:** Environment parameters for experiments.

Experiment Case	Initial Position (m)	Final Position (m)	Virtual Wall Position (m)	Moving Speed (m/s)	Acceleration (m/s^2^)	Deceleration (m/s^2^)
Sudden Human Impedance Change	−0.15	0.15	-	0.3	0.5	−10.0
Virtual Rigid Wall Collision	0.0	0.1	0.05	0.02	0.5	0.5

**Table 4 sensors-21-01899-t004:** Parameters of comparative controllers for experiments.

**Comparison Group**	$M_{d}$ **(kg)**	$D_{d}$ **(Ns/m)**	$M_{d,0}\;(\mathrm{kg})$	$D_{d,0}\;(\mathrm{Ns}/m)$	α
Low Gain Admittance Control	0.2	2	-	-	-
High Gain Admittance Control	2	10	-	-	-
Variable Admittance Control	-	-	0.2	2	10

## Data Availability

This paper did not generate research data to share.

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
