# Peer review of "Variable Admittance Control Based on Human–Robot Collaboration Observer Using Frequency Analysis for Sensitive and Safe Interaction"

_sensors, 2021, doi:10.3390/s21051899_

Round 1

Reviewer 1 Report

The authors in their work consider a crucial topic of admittance controller tuning in human-robot collaboration task. Although the paper is valuable, there are some aspects that should be improved before publication. At first, I can not agree with the general statement characterising admittance control (lines 40 - 42). The problem of this work is that authors analyse a specific type of systems with admittance controller and treat it as the main one in the domain and well known. You should precisely describe and depict your's system structure. Especially the hardware one in the experimental section and general system structure in section 2 (not only the filter in figure 1), should be added. 

The other general problem of the work is that the ideas are mixed with experimental results. I advise to split it. It is visible in section 1, section 2, section 3. Section 4 is only part of the experimental verification because there is a large number of experimental results presented before.

I advise looking to the following works I know that may be valuable for your research. 

T. Winiarski and A. WoĹşniak

Indirect force control development procedure

Robotica, vol. 31, no. 03, pp. 465–478, Apr. 2013

The author proposes an approach to admittance controller development, basing on the frequency methods. In this work, there are also two other aspects essential for you being considered. The first one is an on-line adaptation of the admittance controller parameters, basing on the contact behaviour. The other is the measurement of the inertial forces, occurring in the wrench sensor, and its influence on the admittance controller. If you want to study this particular aspect further, I advise looking to the following paper.

T. Kroger, D. Kubus, and F. M. Wahl. Force and acceleration sensor fusion for compliant manipulation control in 6 degrees of freedom. Advanced Robotics, 21(14):1603–1616, 2007.

Author Response

The authors would thank the reviewer for helpful comments and suggestions on the submitted paper.

Responses to reviewer comments were written in a Word file.

Reviewer 2 Report

The paper seems very interesting in the technical part. However, it needs lots of improvements. I cannot see the novelty since there is not a comparison that highlights the findings. Besides, the paper is written as homework for bachelors. It must improve almost all the figures. A deep English revision is needed. My major concern is the literature, 95% of the cited papers are very low-quality conferences. Instead of this, use magazines, prestigious journals, and transactions.

In the Introduction section, lines 44 to 50 are missing references.

It is recommended to add a conceptual background of conventional techniques in the Introduction section (The introduction does not show state of the art).

The Introduction section does not show the purpose of the article. I suggest that you explain the objective of the paper in detail at the end of the Introduction section. What else does this study contribute? Highlight the importance.

Line 103: HSO definition is missing.

Figure 2 and 3, x-axis: According to the international system of units, the unit of time is s, (avoid using sec).

Add some references to section 2.

Section 4 should be better-named Results.

It is recommended to show in a table the initial conditions of the experiment.

Improve the quality of figures 6, 8, and 11.

The results must be contrasted with other research works.

I suggest adding a nomenclature list.

Grammar: Check the grammar of the text; there are few details (the grammar is generally fine and is easy to read), some are listed below:

  • 93-94:”… That after a period”, the idea is not well-read.
  • “And, ? − 1 is the unit delay”. – you have the preposition “and” after a period.
  • “And it can be confirmed from Figure 5 (b) that the admittance control, which was unstable at human stiffness, becomes asymptotically stable” – you have the preposition “and” after a period.

Author Response

(The authors gave the same response as above.)

Reviewer 3 Report

This paper presents a variable admittance controller to maintain a sensitive and safe human robot interaction. 

I have the following questions and comments:

How do you define sensitive human robot interaction? 

Why did you choose admittance control approach over all possible control methods? 

Why do we need to distinguish between the intended and unintended motions?

The observer used in this paper is a linear one. How did you ensure its robustness?

Figure 5(a) is not explained well. More comments are required.

Are there any similar controller to maintain sensitive and safe human robot interaction. How do you compare your approach with those approaches?

Author Response

(The authors gave the same response as above.)

Round 2

Reviewer 1 Report

The authors applied reviewers remarks. The paper was significantly improved.

Author Response

The authors would thank to the reviewer for helpful comments and suggestions on the submitted paper.

To further improve the paper, we once again entrusted the English correction from EDITAGE, as a company that provides professional, scientific English language editing service (http://www.editage.com).

Reviewer 2 Report

The authors responded to the concerns clearly and concisely. However, I only have certain recommendations for this review.

Section 6. The discussion should be renamed Conclusion.

"Nevertheless, when the operator and the robot are in continuous contact, safety during the physical human–robot interaction is the most important consideration" I recomend you to include te literature regarding security in robotics: 

Adjustable speed drive project for teaching a servo systems course laboratory

An fpga-based open architecture industrial robot controller

Concurrent optimization for selection and control of ac servomotors on the powertrain of industrial robots

Author Response

The authors would thank the reviewer for helpful comments and suggestions on the revised manuscript.

Responses to reviewer comments were written in a Word file.

Reviewer 3 Report

The paper is revised extensively and the comments are addressed well. No more objections, I support its publication.

Author Response

(The authors gave the same response as above.)
